# Diversity of Epibionts Associated with *Lepidochelys olivacea* (Eschscholtz 1829) Sea Turtles Nesting in the Mexican South Pacific

**DOI:** 10.3390/ani11061734

**Published:** 2021-06-10

**Authors:** Brenda Sarahí Ramos-Rivera, Himmer Castro-Mondragon, José Gabriel Kuk-Dzul, Pedro Flores-Rodríguez, Rafael Flores-Garza

**Affiliations:** 1Facultad de Ecología Marina, Universidad Autónoma de Guerrero, Av. Las Palmas No. 20 Fraccionamiento las Playas, Acapulco de Juárez C.P. 39390, Mexico; sarahi.rr19@gmail.com (B.S.R.-R.); himmercm@gmail.com (H.C.-M.); josekuk@gmail.com (J.G.K.-D.); pfloresrodriguez@yahoo.com (P.F.-R.); 2Dirección de Cátedras-CONACYT, Consejo Nacional de Ciencia y Tecnología (CONACYT), Ciudad de México C.P. 03940, Mexico

**Keywords:** epibionts, interspecific relationships, *Lepidochelys olivacea*, Mexico, sea turtle, South Pacific

## Abstract

**Simple Summary:**

Epibionts are organisms that live or grow attached to other living beings, and sea turtles can be suitable habitat for these organisms because they provide a large and diverse substrate. They usually have interspecific relationships of the commensal type; however, some species become parasitic and may cause severe damage, mainly in soft areas. Epibionts provide us with information on the migratory habits of sea turtles and can indicate health status. There are several studies on epibionts and their relationships with sea turtles; however, it is essential to expand research to increase the knowledge that will allow us to comprehend these relationships and their implications. In this study, we analyze the richness, abundance, diversity, prevalence, body distribution, and interspecific relationships of epibionts with *Lepidochelys olivacea* turtles nesting in the Mexican South Pacific, relate turtle size with the presence of epibionts, characterize the body distribution of epibionts, determine the affinity in species composition, and document the interspecific relationships.

**Abstract:**

The present study contributes to the knowledge of epibionts recorded on sea turtles that nested on a beach in the South Pacific of Mexico. A total of 125 *Lepidochelys olivacea* turtles nested on Llano Real beach, Guerrero, Mexico, were examined. We collected 450 conspicuous organisms from 8 species from 43 turtles. The corresponding data analysis was carried out to obtain the relative abundance, the relationship between turtle sizes and the presence of organisms, the similarity of species between the sampling months, and the interspecific relationships between the epibionts and the turtles observed. *Chelonibia testudinaria* was the most abundant species, while *Remora remora* was the least abundant species. The turtles were divided into six body sections, with the greatest abundance of these organisms located in the head–neck section of turtles, and there was a significant difference in the size of the turtles that presented epibionts and those that did not. *C. testudinaria* showed greater similarity between sampling months, and the interspecific relationships recorded were commensalism, parasitism, amensalism, and protocooperation. This research contributes the first record of epibionts in *L. olivacea* nesting in Guerrero, Mexico.

## 1. Introduction

Epibionts, being organisms that grow and live attached to other species, are useful to those seeking knowledge of the biology and ecology of the living being that serves as a substrate in this association [1]. Many studies have created lists of epibionts that have been identified on different hosts [2,3]. The body surface of sea turtles is often used by epibiont fauna as a settlement substrate and as a means of dispersal and food procurement [4]. Analysis of epibionts may provide information on sea turtle biology and ecology, indicating types of environments passed through, migration times and depth, regional occurrence, habitat use, health, seasonality, behavior, gender-based patterns, and signs of climate change [5,6].

The epibionts of mobile organisms also encounter unfavorable conditions, such as morphological and physiological changes of the basibiont, or friction with other species [7]. They can also be eaten by its basibiont’s predators [8] and suffer abrupt environmental changes, especially those epibionts that live on organisms that have large movements in distance and depth. One of the most extreme examples is the case of the epibionts that live on the carapace of the loggerhead turtle (*Caretta caretta*), since this basibiont passes through coastal, oceanic, and even terrestrial environments in tropical and subtropical areas [9].

Sea turtles are possibly the marine species with the most diverse epibiont communities due to the great variability in movement patterns and feeding preferences among individuals. Therefore, sea turtles and their epibionts are useful as biological models to investigate factors influencing interspecific variation in epibiont community structure [10,11,12]. Comprehensive taxonomic analyses of epibiotic fauna may reveal clustering patterns and distinguish groups of sea turtles [13]. Further, because sea turtles nest on tropical beaches around the world, they are helpful for examining how epibionts respond to hosts leaving the water.

Epibionts have been considered indicators of health in sea turtles. In the first instance the intraspecific relationship therebetween is considered commensalism, but this relationship may change to amensalism or even parasitism depending on the number of organisms and the effects of these on a single turtle [14,15]. In *Lepidochelys olivacea* (Eschscholtz 1829), areas most affected by epibiosis are the carapace, flippers, head and neck [16], and the diversity of species is lower than in others, such as the loggerhead turtle (*Caretta caretta*) and hawksbill turtle (*Eretmochelys imbricata*), of which more than 100 epibiont species have been described [17]. On the Mexican coasts, 21 species of conspicuous epibiont fauna have been reported for *L. olivacea* [18] and studies on epibionts in *L. olivacea* in the Mexican Pacific have been conducted by Hernández and Valadez [19], Gámez et al. [16], Ayala and González [20], and Frick et al. [21]. The aims of this study were to: (1) determine the richness, abundance, diversity, and prevalence of epibionts in *L. olivacea* turtles nesting in the Mexican South Pacific; (2) relate turtle size with the presence of epibionts; (3) characterize the distribution of epibionts on turtles’ bodies; (4) determine the affinity in species composition; and (5) document interspecific relationships between *Lepidochelys olivacea* and its epibionts.

## 2. Materials and Methods

The fieldwork was carried out in Guerrero, Mexico, on the beach Llano Real where the Sea Turtle Conservation and Protection Center is located (coordinates 17°04′00.4′′ N, 100°26′56.8′′ W). This center is administered by the Universidad Autónoma de Guerrero (Figure 1).

Night tours were conducted from July to November 2017 to locate nesting turtles, and once the turtle was found, we examined it carefully. The same two collectors conducted the epibiont assessments, for which the turtle’s body was divided into six sections (head/neck, front flippers, rear flippers, carapace, plastron, and cloaca), following almost the same standardization suggested by Lazo-Wasem et al. [2]. For each body section, the conspicuous fauna were collected and deposited in labeled plastic vessels. After collection, turtle curved carapace length (CCL) and curved carapace width (CCW) biometrics were recorded. The conspicuous epibionts were preserved in alcohol (70%) and transferred to the laboratory for identification and quantification. Identification of epibiont species was carried out using specialized literature [22,23,24], and nomenclature was reviewed using the World Register of Marine Species (WoRMS) site.

Epibiont species richness was determined using the Margalef diversity index. Relative abundance was estimated by dividing the individuals of each epibiont species between the total numbers of organisms collected, and expressed as a percentage. Diversity was estimated with the Shannon–Wiener index.

To determine prevalence, the number of turtles with epibionts was divided by the total number of turtles analyzed and expressed as a percentage. We used the Mann–Whitney U test to test differences between sizes (CCW) of turtles that showed epibionts and those that did not. Additionally, we determined the body distribution of epibionts, species richness, abundance, and frequency of occurrence per body section. To test affinity in epibiont species composition between months, a dendrogram was formed using the Bray–Curtis dissimilarity (transforming the data to square root and grouping data to determine the similarity between sampling months). Specialized literature was consulted to determine the interspecific relationship between epibionts and turtles [16,19,21]. In addition, to define interspecific relationships, we recorded direct observations of species of epibiont, abundance per turtle, and the type of interaction at the time of collection.

## 3. Results

We examined a total of 125 females of *L. olivacea* nested on the beach (Sea Turtle Conservation and Protection Center). Conspicuous epibiont fauna were observed in 43 turtles, from which 450 specimens were collected, and 3 phyla, 5 families, 8 genera, and 8 species were identified (Figure 2). The prevalence of epibionts was 34.4%, the species richness according to the Margalef index was 1.146, and the estimated Shannon–Wiener diversity index was 2.198 bits/individual.

The epibiont with the highest abundance was the barnacle *Chelonibia testudinaria* (195 specimens), followed by the annelid *Ozobranchus branchiatus* (110 specimens). The species with the lowest abundance and prevalence was *Remora remora*, the only vertebrate present within the epibiont group; this species was collected in two females (Table 1).

Mean estimated CCW in turtles in which no epibionts were observed was 66.8 ± cm (σ: 8.29), for turtles where epibionts were found, the mean estimated CCW was 69.3 ± cm (σ: 2.95). According to the Mann–Whitney U test, the existence of a significant statistical difference between turtle CCW averages was estimated (U = 1334.0, *p* = 0.025).

For the body distribution of epibionts, the body section with the highest species richness was the carapace, where seven of the eight species reported in this research were found, followed by the head–neck and front flippers sections, where six species were recorded. However, the highest abundance of epibionts was observed in the head–neck (291 specimens), followed by the carapace section (117 specimens, Table 2). No epibionts were observed in the cloaca. The species *C. testudinaria* was presented as the one with the broadest body distribution since it was observed in five of the six body regions into which the turtle was divided. It was also the most abundant, presenting high abundances in the body of the turtle.

Concerning the affinity in species composition, the Bray–Curtis dissimilarity indicated the formation of six groups according to clustering analysis of the species and abundance of epibionts collected from July to November 2017 (Figure 3). Groups e and f showed >50% similarity; group e included species collected during July and August, while group f were species collected during September and November. Group d had only one species collected in July and one in November (40% similarity). Group c had organisms collected in September, October, and November (>30% similarity). Group b obtained <20% similarity, and finally group a was represented by one species collected only in August, obtaining 0% similarity with the other months (Table 3).

Based on the specialized literature [16,19,21,25] it was recognized that *C. testudinaria, L. hilli, C. virgatum, P. hexastylos,* and *S. elegans* were related to commensalism; *S. muricata* to amensalism; *O. branchiatus* to parasitism; and *R. remora* to protocooperation, with observed sea turtles.

## 4. Discussion

In Mexico, 21 species of conspicuous epibionts have been reported for *L. olivacea* [18]. Eighteen epibiont taxa have been reported from turtles [2,16,19,26,27,28], observed at various sites in ecoregion No. 17 [29], where the state of Guerrero is located (Table 4). In the present study, we documented only 38% of the epibiont fauna reported for the Mexican Republic, and 44% of that reported for ecoregion 17 were found in the turtle population analyzed.

The prevalence of epibionts in the population studied was low and given that epibionts have been considered as indicators of sea turtle health, since a sick turtle increases the probability of a greater load of epibionts [14], the low prevalence of epibionts is a good health indicator of the population of nesting turtles in the study area. In addition, the low species richness, the low estimated diversity index, and the abundance recorded strengthen the aforementioned assumption; however, it is necessary to test this hypothesis, given that the diversity of epibionts in *L. olivacea* has been reported to be lower when compared to other sea turtle species [17].

According to Márquez [30] and Zug [31], in *L. olivacea*, as age increases, the ratio of width to length increases, so that older turtles will be wider. The significant difference estimated in CCW size between turtles with and without epibionts in the present study is evidence that strengthens the assumption that the older the turtle, the greater the epibiont load.

Of the species reported by this research, only *S. muricata* had not been found in ecoregion 17; however, there are reports [21,32] of this species in the state of Sinaloa (Table 4).

*C. testudinaria* has been considered an “obligate” epibiont species because it has been found in six species of sea turtles [16,32]. This species secretes a substance that allows it to adhere to areas with rigid substrate and has been found on the heads, noses, and carapaces of sea turtles [3]. In addition to being found in the body sections, it was also observed in the nails in the population studied. Furthermore, this study reports this species as the most widely distributed among turtle body sections and the most abundant in the site in ecoregion 17 [2,26,28]. Further, in addition to being found in 5 of 6 turtle body sections, it was also observed on the claws on the front flippers.

The annelid *O. branchiatus* is hematophagous [33] and was presented as the species that occupied the second place in relative abundance, coinciding with what Gámez et al. [15] reported; however, Hernández-Vázquez and Valadez-González [18] reported it as the most abundant epibiont in *L. olivacea*. This species was found mainly on the neck and front flippers in the population studied and was also observed moving on the carapace.

The species *L. hilli* and *C. virgatum* belong to the Lepadidae family, which occupies habitat associated with floating objects and secretes an adhesive substance through the peduncle, so they can be found on any part of the turtle, whether hard or soft, and their presence on turtles implies that they frequent shallow depths close to the surface layers of the sea [34,35]. These two species were found in four of the turtle specimens analyzed, but the greatest abundance was recorded in two: the first had a piece of rope entangled in a front flipper, and the second was observed with damage to the carapace, both considerably injured and weakened, so it is assumed that their swimming was slow, and they spent more time floating on the surface, facilitating the adherence of these organisms.

It was reported that members of the genus *Stomatolepas* are common commensals of sea turtles; these small barnacles often attach themselves to the soft skin areas of their hosts and are frequently located in the neck area [23], which coincides with what was observed in the field for *S. elegans.* For other barnacles of the same family, such as *S. muricata*, it has been described that it has a screw shape and finely serrated ornamentations with which it pierces soft areas such as the neck and fins and develops inside the skin [21]. Moreover, for *P. hexastylos* it was indicated that this species adheres more easily to hard substrates by using a membranous base with grooves that allow it to attach to turtles [36]. Our study corroborates the abovementioned, due to finding *S. muricata* located in parts of the neck and front flippers inside the skin and *P. hexastylos* with greater abundance near the mouth and nose of the turtles. The vertebrate *R. remora* was the least abundant epibiont and was found on the turtle carapace; it has also been reported as the least abundant, and has been found in turtle nests, so presumably, these were attached to the plastron [19].

It has been reported that the turtle’s neck, being soft-skinned, is more vulnerable to colonization by epibionts [22]. This research corroborates this since the highest abundance of epibionts was recorded in the head/neck section.

In the plastron section, a low abundance of organisms was observed because it is likely that the epibionts are detached from the turtle when it comes out to lay eggs, due to sand friction. The only section where no attached organisms were found was the cloaca; in other studies, coprophagous crabs of the species *Planes major* have been found [2].

Of the eight species collected, it can be seen that some had affinity among them during the collection period: *C. testudinaria* was the one that occurred in greater abundance and therefore obtained greater similarity between months. From September, October, and November onwards, different species were collected with >50% similarity, unlike July and August, where the species affinity was <20%. The only species for which no similarity was obtained was *R. remora* since it was only collected in August. This indicates that turtles with greater species similarity may have the same migration routes, feeding, and breeding areas and that the turtles with greater size were those with greater epibiont similarity.

The importance of studying symbiotic relationships between epibionts and sea turtles has been pointed out as it aids in determining the effect that epibionts have on their hosts [37]. Species such as *C. testudinaria, P. hexastylos, L. hilli, C. virgatum,* and *S. elegans* found in this research were cataloged as commensal species, similar to the conclusion of Casale et al. [38] and Pinou et al. [24], who reported that these species only use the turtle as a substrate and adhere to it from their cypris larval stage, until completing their life cycle. Based on the size, weight, and abundance of the species mentioned above, it is considered that they are not harmful to the turtles analyzed. It was reported that *R. remora* establishes a proto-cooperative relationship; this interaction presents advantages for the remora that include transportation without energy expenditure, obtaining food, and mating opportunities, in addition to “cleaning” the turtles from other epibionts [39]. The leech *O. branchiatus* was the only epibiont that was considered a strict ectoparasite. Oceguera-Figueroa and León-Règagnon [33] report that this species feeds directly on the turtle and can cause considerable damage, as it weakens it by consuming its blood. The barnacle *S. muricata* does not feed on the turtle, but it does cause considerable damage because it pierces the soft parts, such as the flippers and neck. Ross and Frick [40] report that this species has fine lateral serrated ornamentation, which it uses to hold onto turtles and with which it opens the skin; even when the turtle produces a layer of fat around the organism, it breaks through this to be able to develop. Zardus [25] considered that the relationship established between *S. muricata* and sea turtles is amensalism, which comports with our observations, including that the turtles did not suffer significant damage in this research from those organisms.

## 5. Conclusions

We report the first record of epibionts in *L. olivacea* turtles nesting in Guerrero, Mexico. Other reports confirm that the diversity of epibionts in *L. olivacea* females is low compared to other sea turtle species. In the study area, only 44% of the epibiont species reported in different sites of ecoregion 17 were found to be establishing some type of interspecific relationship with the nesting females of *L. olivacea*.

We report for the first time the presence of the species *S. muricata* for ecoregion 17 (Mexican Transitional Pacific) as an epibiont on *L. olivacea*. The presence of *C. testudinaria* as an “obligate” epibiont in the population studied was corroborated, but it was also the most abundant and widely distributed species in the turtle’s body in the study area. The head/neck section was the one that was inhabited mainly by epibionts without significant effects of their presence in the turtle population.

Due to the similarity of epibionts recorded between sampling months, it was considered that the turtles nested at the study site might have different feeding and breeding areas. The epibionts recorded in this study have interspecific relationships of commensalism, parasitism, amensalism, and protocooperation with sea turtles.

## Figures and Tables

**Figure 1 animals-11-01734-f001:**
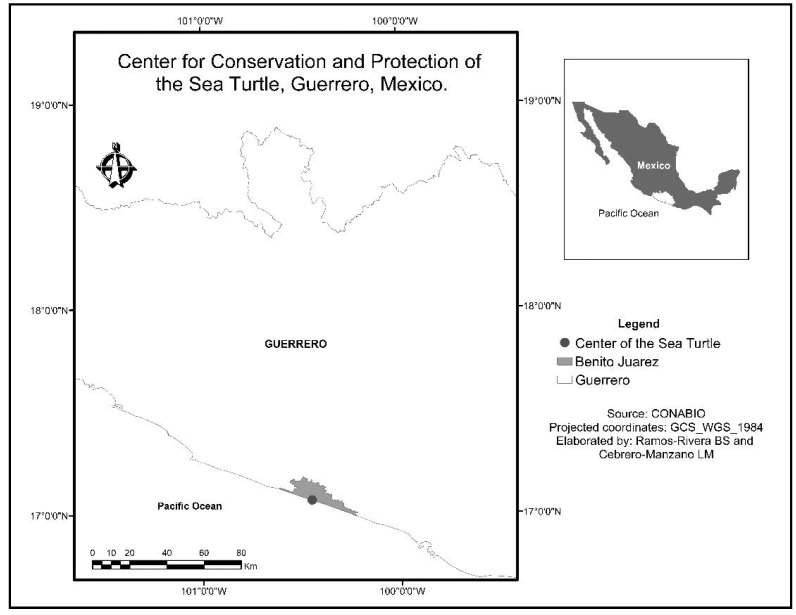
Location of the study site in Guerrero, Mexico during 2017.

**Figure 2 animals-11-01734-f002:**
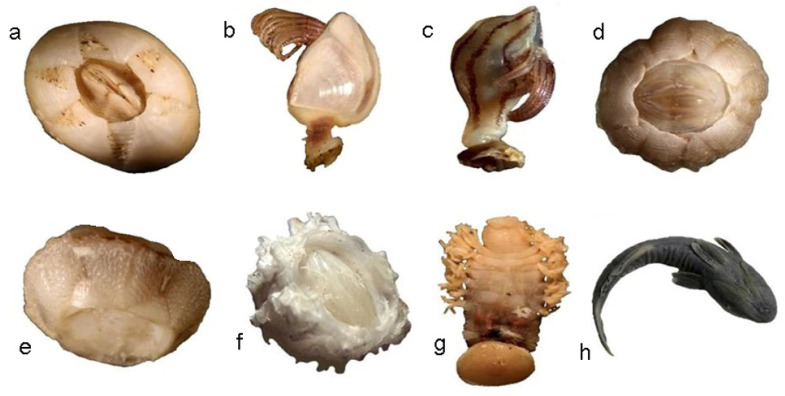
Epibionts on *Lepidochelys olivacea* collected during the season July to November 2017 in Guerrero, Mexico. (**a**) *Chelonibia testudinaria*; (**b**) *Lepas hilli*; (**c**) *Conchoderma virgatum*; (**d**) *Platylepas hexastylos*; (**e**) *Stomatolepas elegans*; (**f**) *Stephanolepas muricata*; (**g**) *Ozobranchus branchiatus*; (**h**) *Remora remora.*

**Figure 3 animals-11-01734-f003:**
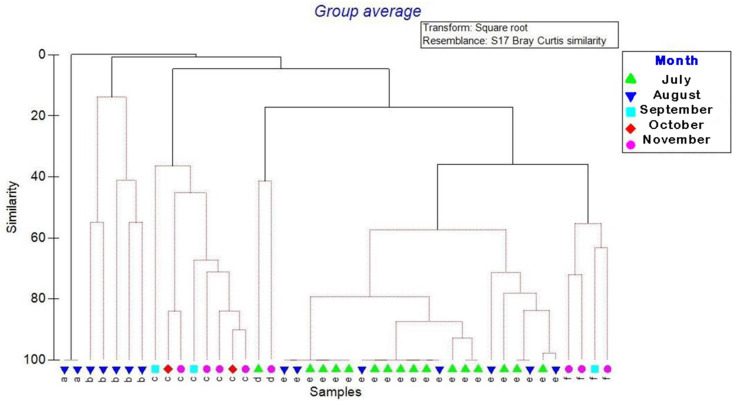
The similarity of epibiotic species composition in *Lepidochelys olivacea* per sampling month in Guerrero, Mexico.

**Table 1 animals-11-01734-t001:** Richness, abundance, and prevalence of epibionts in turtles that nested on the beach where the Center for Conservation and Protection of the Marine Turtle is located, Guerrero, Mexico.

Phylum	Family	Species	TA	RA (%)	NT	P (%)
Arthropoda	Chelonibiidae	*Chelonibia testudinaria* *(Linnaeus,1758)*	195	43.3	28	22.4
	Lepadidae	*Lepas hilli* *(Leach,1818)*	72	16.0	4	3.2
		*Conchoderma virgatum* *(Spengler, 1790)*	13	2.9	4	3.2
	Platylepadidae	*Platylepas hexastylos* *(Fabricius, 1798)*	28	6.2	7	5.6
		*Stomatolepas elegans* *(Costa, 1838)*	19	4.2	5	4.0
		*Stephanolepas muricata* *(Fischer, 1886)*	11	2.4	3	2.4
Annelida	Ozobranchidae	*Ozobranchus branchiatus* *(Menzies,1791)*	110	24.4	10	8.0
Chordata	Echeneidae	*Remora remora (Linnaeus,1758)*	2	0.4	2	1.6

Note: TA = total abundance; RA = relative abundance; NT = number of turtles where the epibiont was found; P = prevalence by epibiont species.

**Table 2 animals-11-01734-t002:** Distribution of epibionts by body section in Guerrero, México.

Species	The Abundance of Epibionts per Body Section						FO
	1	2	3	4	5	6	
*Chelonibia testudinaria*	138	1	11	43	2	0	5/6
*Conchoderma virgatum*	8	4	0	1	0	0	3/6
*Lepas hilli*	0	8	0	64	0	0	2/6
*Ozobranchus branchiatus*	99	8	0	3	0	0	3/6
*Platylepas hexastylos*	19	4	0	2	3	0	4/6
*Remora remora*	0	0	0	2	0	0	1/6
*Stephanolepas muricata*	10	1	0	0	0	0	2/6
*Stomatolepas elegans*	17	0	0	2	0	0	2/6
Total of species for corporal section	6	6	1	7	2	0	
Total specimens for corporal section	291	26	11	117	5	0	

Note: (1) head–neck; (2) front flipper; (3) back flipper; (4) carapace; (5) plastron; (6) cloaca; FO = frequency of occurrence by body sections.

**Table 3 animals-11-01734-t003:** Affinity in the composition of epibiont species between the months.

Groups	Species	Months	Similarity between Months
a	*R. remora*	August	0%
b	*Conchoderma virgatum and Lepas hilli*	August	<20%
c	*Ozobranchus branchiatus and Platylepas hexastylos*	September, October and November	>30%
d	*Stephanolepas muricata*	July and November	>40%
e	*Chelonibia testudinaria*	July and August	>50%
f	*Chelonibia testudinaria y Stomatolepas elegans*.	September and November	>50%

**Table 4 animals-11-01734-t004:** List of epibiont fauna in *L. olivacea* in sites located in ecoregion 17 (Mexican Transitional Pacific).

Species	Sites of 17 Ecoregion						
	a	b	c	d	e	f	g
*Chelonibia testudinaria*	X	X		X		X	X
*Conchoderma virgatum*	X	X				X	X
*Lepas hilli*						X	X
*Ozobranchus branchiatus*	X	X				X	X
*Platylepas hexastylos*	X					X	X
*Remora remora*	X					X	X
*Stephanolepas muricata*							X
*Stomatolepas praegustator*			X				
*Stomatolepas elegans*						X	X
*Planes cyaneus*	X						
*Planes major*						X	
*Dulichia sp.*							
*Caprella sp.*		X					
*Balaenophilus umigamecolus*					X		
*Balaenophilus manatorum*			X			X	
*Podocerus chelonophilus*						X	
Gammaridae	X						
Actinaria						X	

(a) Hernández-Vázquez & Valadez-González [19]; (b) Gámez et al. [16]; (c) Lazo-Wasen et al. [27]; (d) Gámez et al. [28]; (e) Suárez-Morales et al. [26]; (f) Lazo-Wasem et al. [2]; (g) present research.

## Data Availability

The data presented in this study are available on request from the corresponding authors.

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
