# Peer review of "Diversity of Epibionts Associated with Lepidochelys olivacea (Eschscholtz 1829) Sea Turtles Nesting in the Mexican South Pacific"

_animals, 2021, doi:10.3390/ani11061734_

Round 1

Reviewer 1 Report

Review

Paper title: Diversity of epibionts associated with Lepidochelys olivacea (Eschscholtz 1829), sea turtles nesting in the Mexican South Pacific.

Sea turtles play an important role in the areas where they occur as ecosystem engineers maintaining the health of seagrass beds and coral reefs. They provide habitats for marine organisms, aid in maintaining balanced marine food webs, and promote nutrient cycling from marine to terrestrial ecosystems. They are considered important indicators of the health of marine ecosystems. Epibiotic communities of sea turtles are useful indicators of many aspects of the biology of these hosts including health, feeding and migrations. In this paper, the authors for the first time described the epibiont fauna of olive ridley (Lepidochelys olivacea) in the Mexican South Pacific. They found 8 species including commensals and parasites. According to an analysis of seasonal patterns in epibiotic communities, the authors concluded the turtles nested at the study site may have different feeding and breeding areas. This result may have important implications for further studies and management of the turtle population.

All these reasons explain the relevance of the paper by Brenda Sarahí Ramos-Rivera and co-authors submitted to "Animals".

General scores.

The data presented by the authors are original and significant. All conclusions are justified and supported by the results. The study is correctly designed and technically sounds. In general, the statistical analyses are performed with good technical standards. We authors conducted careful work which will attract the attention of a wide range of specialists focused on the biology of sea turtles, managers and ecologists.

Specific comments.

L 2. " Lepidochelys  olivacea" should be italicized.

L 19. Change “their relationship” to “their relationships”

L 20. Change “this relationship and its” to “these relationships and their”

L 26. Change “[1]turtles” to “sea turtles”

L 27. Change “L. olivacea” to “Lepidochelys  olivacea

L 68. Change “L. olivacea” to “Lepidochelys  olivacea (Eschscholtz 1829)”

L 68. A short description of this host sea turtle is required (distribution, ecology, role in the ecosystem, etc.).

L 74. Change “Hernández and Valadez (1998) [18], Gámez et al. (2006)” to “Hernández and Valadez [18], Gámez et al.”

L 75. Change “González (2006) [19], Frick et al. (2011)” to “González [19], Frick et al.”

L 115. " L. olivacea" should be italicized.

L 121. Change “L. olivacea” to “Lepidochelys  olivacea

L 125. Change “C. testudinaria” to “Chelonibia  testudinaria

L 126. Change “O. branchiatus” to “Ozobranchus branchiatus

L 129. Change “R.  remora” to “Remora remora

Table 1, column 3. Change “Specie” to “Species”

L 146. Change “turtle.” to “turtle”

Table 2, column 1. Change “seccion” to “section” (twice). Please, use full species names for the epibionts.

Table 3, column 1. Change “Grups” to “Groups”

Table 3, column 2. Change “Specie” to “Species”. Please, use full species names for the epibionts. Change “y “ to “and”

L 163. Change “C. testudinaria L. hilli C. virgatum P. hexastylos S. praegustator” to “C. testudinaria, L. hilli, C. virgatum, P. hexastylos and S. praegustator”

L 163-165. “It was determined…” means that the authors studied the relationships between ebipionts and their hosts, but they didn't do it. For this reason, I suggest deleting this paragraph.

L 170-171, 178-179. The authors should provide explanations for these results. Why they registered lower epibiont diversity and prevalence in the study area?

L 180. Change “Márquez (1996)” to “Márquez”

L 196. Change “(2006) [15]” to “[15]” , “(1998) [18]” to “[18]”

L 239. Change “symbiosis relationships” to “symbiotic relationships”

L 241. The citation “(Mota and Lara 2014)” is presented in the text but absent in the list of references.

L 251. Change “(2014) [29]” to “[29]”

L 254. Change “Frick (2011)” to “Frick”

L 274-275 The sentence “We report the first…” should be placed at the beginning of the “Conclusions” section.

L 301. " Posidonia Oceanica " should be italicized.

L 301-302. Change “Journal of  Marine Ecology” to “Marine Ecology”

L 308. Change “Journal of Biology Letters” to “Biology Letters”

L 316. Change “Journal of Marine Turtle Newsletter” to “Marine Turtle Newsletter”

L 321. Change “Journal of Veterinaria Mexico” to “Veterinaria Mexico”

L 335. Change “Journal of Marine Turtle Newsletter” to “Marine Turtle Newsletter”

L 339. " Balaenophilus Umimegacolus" should be italicized.

L 340. Change “Journal of Crustaceana” to “Crustaceana”

L 342. " Balaenophilus  Umigamecolus " should be italicized.

L 359. Missing year of publication.

L 363. Change “Journal of Crustaceana” to “Crustaceana”

L 368. Change “Journal of Zootaxa” to “Zootaxa”

Author Response

We are grateful for your good comments and for taking the time to review our manuscript.

We responded promptly to the suggestions that you mentioned to us.

Kinds regards.

Reviewer 2 Report

Although I feel that any sampling of sea turtle epibionts is critical and valuable, I have concerns about the lack of citation rigor of this paper.  Meaning the citations do not align with the information presented and the authors text.  It's as if the paper went through revisions but the authors never double checked to make sure the citations agreed with what they were saying.  A few examples, although you include reference 6, this standardization isn't referred to in the methodology - but rather you cite 21 and 22 which are inappropriate and have little to do with sampling methodology.  Also, you include a more recent methods and standards citation in the references but never use it. In the methodology you never tell us if this is an exhaustive collection of epibionts, in fact you say nothing about standardizing how or who - yet you provide results that talk about regional abundance.  I find this unethical and unscientific - especially when you then mention concluding ideas in lines 138-146 - yet we still have no idea how you standardized sampling?  Line 163 states "it was determined that XXXXXX" - how was all this determined? Did you use a published key? Literature? what did you use to determine species identity?  Again, this lends itself to my concerns with quality control of the study.  On Table 4 you mention Stomatolepas praegustator and S. elegans, and highlight that you have found S. muricata.  However, you never mention Pinou et al (2013) on Stomatolepas 

http://dx.doi.org/10.1080/00222933.2013.798701

That highlighted the plasticity and enormous diversity of this genus.  The paper clearly showed that S. elegans and S. praegustator are likely the same species, and the shell plasticity likely a result of their attachment and positioning.  Your conclusion that affirms low diversity on L. olivacea in the Pacific is interesting and good but overall the paper needs to be carefully revisited for content, and the breadth of literature on LO epibionts from the Pacific integrated or else this is little more than another list.  How does your snapshot build our knowledge, what patterns do you see regarding movement.  Where the turtles tagged?  Any of them?  Mexico has a well-documented tagging program as does Neighboring countries, what is the identity of the turtles you sampled from? Did you check?  In fact, that's the piece I find amazing, where are all the tagged sea turtles in Mexico?  This may help you support some of your statements regarding epibionts reflecting sea turtle migration - which I see no effort in contributing to with this paper? Maybe the natural history of these epibionts can help in this regard?  Figure 3 to me is strange and meaningless if we cannot verify the reproducibility of your sampling method, as what you report here may be more about who sampled and what they favored to collect, or what was convenient to collect.  More about sampling rigor must be included in this paper.

Author Response

Point 1

Although I feel that any sampling of sea turtle epibionts is critical and valuable, I have concerns about the lack of citation rigor of this paper.  Meaning the citations do not align with the information presented and the authors text.  It's as if the paper went through revisions but the authors never double checked to make sure the citations agreed with what they were saying.

Response 1: Thank you for your comments, We modify the references correctly.

A few examples, although you include reference 6, this standardization isn't referred to in the methodology - but rather you cite 21 and 22 which are inappropriate and have little to do with sampling methodology.  Also, you include a more recent methods and standards citation in the references but never use it. In the methodology you never tell us if this is an exhaustive collection of epibionts, in fact you say nothing about standardizing how or who - yet you provide results that talk about regional abundance.  I find this unethical and unscientific - especially when you then mention concluding ideas in lines 138-146 - yet we still have no idea how you standardized sampling?

Response 2: The standardization of reference # 6 was not used because that publication came out 2 years after having made our collection, however we relied a little on reference # 2 to "standardize" the sections in which we divided the turtles and added the missing part in that section.

Line 163 states "it was determined that XXXXXX" - how was all this determined? Did you use a published key? Literature? what did you use to determine species identity?  Again, this lends itself to my concerns with quality control of the study. 

Response 3: We make the modification corresponding to your observation.

On Table 4 you mention Stomatolepas praegustator and S. elegans, and highlight that you have found S. muricata.  However, you never mention Pinou et al (2013) on Stomatolepas

http://dx.doi.org/10.1080/00222933.2013.798701

That highlighted the plasticity and enormous diversity of this genus.  The paper clearly showed that S. elegans and S. praegustator are likely the same species, and the shell plasticity likely a result of their attachment and positioning.

Response 4: We correctly modify your observation and include the recommended reference.

 Your conclusion that affirms low diversity on L. olivacea in the Pacific is interesting and good but overall the paper needs to be carefully revisited for content, and the breadth of literature on LO epibionts from the Pacific integrated or else this is little more than another list.  How does your snapshot build our knowledge, what patterns do you see regarding movement.  Where the turtles tagged?  Any of them?  Mexico has a well-documented tagging program as does Neighboring countries, what is the identity of the turtles you sampled from? Did you check?  In fact, that's the piece I find amazing, where are all the tagged sea turtles in Mexico?  This may help you support some of your statements regarding epibionts reflecting sea turtle migration - which I see no effort in contributing to with this paper? Maybe the natural history of these epibionts can help in this regard? 

Response 5: Unfortunately in our research we were not able to mark the observed turtles, besides that it was not one of our objectives, we did not have enough resources to acquire the metal plates.

Figure 3 to me is strange and meaningless if we cannot verify the reproducibility of your sampling method, as what you report here may be more about who sampled and what they favored to collect, or what was convenient to collect.  More about sampling rigor must be included in this paper.

Response 6:

We are grateful for your comments and for taking the time to review our manuscript.

We responded promptly to the suggestions that you mentioned to us.

Kinds regards.

Reviewer 3 Report

Dear authors,

Your study is both interesting and valuable but some issues need clarification and I am recommending it for significant revision before it can be accepted for publication. Perhaps the most significant aspect of your study is your novel analysis using the Bray-Curtis index to examine epibiont community composition over time. Yours is one of the few studies to examine changes in epibionts over time and it is a real strength that needs to be highlighted more. I like the approach of your analysis which offers a new perspective for epibiont studies and I have included some comments on some ways you may want to interpret your data.

One major point that needs clarification is whether you collected epibionts from any turtles that were recaptured during the period. If so, how was the data for recaptures handled? I have provided heavy editing of the manuscript to help you in revising in an attached Word version of your manuscript where my edits and comments can be seen using the “track changes” feature in Word.

I point out a few general items below for you to consider in your revision:

1) Discussing barnacles as parasites is problematic. They do in some instances harm their hosts but they are not parasites in the traditional sense of taking nutrition from their hosts. I think it is best to describe them as mostly harmless commensals but in some cases or with some species they do cause harm and just leave it at that. Mutualism is a +/+ relationship, commensalism is a +/0 relationship, amensalism is a 0/- (which does not correctly describe the relationship of epibionts to turtles since all epibionts receive some kind of benefit).

2) You should review and cite the follwoing recent article for your revised version:

Zardus, J.D. (2021). A global synthesis of the correspondence between epizoic barnacles and their sea turtle hosts. Integrative Organismal Biology, 3, https://doi.org/10.1093/iob/obab002

3) A comment about style. When you have data or information presented in a table or plot you should not repeat it again in the text. You can refer to and discuss general trends and perhaps even present average values as long as they are not already given in the table or plot.

4) Be sure to go over my comments in the attached document regarding the dendrogram and how it could be improved. This I think is the most important aspect of your study.

5) The species of Stomatolepas you list is S. elegans. The species Stomatolepas praegustator has only been described from the mouth of sea turtles, Stomatolepas on the necks of olive ridleys are the species S. elegans (see Robinson, N.J., Lazo-Wasem, E., Butler, B.O., Lazo-Wasem, E.A., Zardus, J.D. & Pinou, T. (2019). Spatial distribution of epibionts on olive ridley sea turtles at playa ostional, costa rica. PLoS ONE, 14, e0218838).

Other studies have also found that Stomatolepas praegustator is genetically identical to S. elegans so the two are really the same species and the name S. elegans has priority (see Pinou, T., Lazo-Wasem, E.A., Dion, K. & Zardus, J.D. (2013). Six degrees of separation in barnacles? Assessing genetic variability in the sea-turtle epibiont stomatolepas elegans (costa) among turtles, beaches, and oceans. Journal of Natural History, 47, 2193-2212).

6) On line 216 you discuss Stephanolepas muricata and mention it was found in the neck. Is that right? This is not common so it is important to confirm. This barnacle is usually found only in the flipper.

7) On line 255 you discuss how Stephanolepas invades the skin and gets surrounded by a layer of fat (??) and you mention it affecting turtle eggs (??). I think this is in error. Please clarify or delete.

8) In the discussion you could add more discussion regarding succession in epibiont communities on turtles since little has been studied. Below are the only two references that I know of on the topic.

Frick, M.G., Williams, K.L., Veljacic, D.C., Jackson, J.A. & Knight, S.E. (2002). Epibiont community succession on nesting loggerhead sea turtles, Caretta caretta, from Georgia, USA. In: Proceedings of the 20th Annual Symposium on Sea Turtle Biology and Conservation (eds. A. Mosier, A. Foley and B. Brost), pp. 281-282. NOAA technical memorandum NMFS-SEFSC-477.

Nolte, C.R., Pfaff, M.C., De Lecea, A.M., Le Gouvello, D. & Nel, R. (2020). Stable isotopes and epibiont communities reveal foraging habitats of nesting loggerhead turtles in the south West Indian Ocean. Marine Biology, 167, article 162.

Author Response

Point 1

Your study is both interesting and valuable but some issues need clarification and I am recommending it for significant revision before it can be accepted for publication. Perhaps the most significant aspect of your study is your novel analysis using the Bray-Curtis index to examine epibiont community composition over time. Yours is one of the few studies to examine changes in epibionts over time and it is a real strength that needs to be highlighted more. I like the approach of your analysis which offers a new perspective for epibiont studies and I have included some comments on some ways you may want to interpret your data.

Response 1: Thank you for your goods comments.

Point 2.

One major point that needs clarification is whether you collected epibionts from any turtles that were recaptured during the period. If so, how was the data for recaptures handled?

Response 2:  Unfortunately in our research we were not able to mark the observed turtles, besides that it was not one of our objectives, we did not have enough resources to acquire the metal tags.

I have provided heavy editing of the manuscript to help you in revising in an attached Word version of your manuscript where my edits and comments can be seen using the “track changes” feature in Word.

I point out a few general items below for you to consider in your revision:

1) Discussing barnacles as parasites is problematic. They do in some instances harm their hosts but they are not parasites in the traditional sense of taking nutrition from their hosts. I think it is best to describe them as mostly harmless commensals but in some cases or with some species they do cause harm and just leave it at that. Mutualism is a +/+ relationship, commensalism is a +/0 relationship, amensalism is a 0/- (which does not correctly describe the relationship of epibionts to turtles since all epibionts receive some kind of benefit).

2) You should review and cite the follwoing recent article for your revised version:

Zardus, J.D. (2021). A global synthesis of the correspondence between epizoic barnacles and their sea turtle hosts. Integrative Organismal Biology3, https://doi.org/10.1093/iob/obab002

Response 3:  We promptly addressed this observation and the recommended reference was added to the document.

3) A comment about style. When you have data or information presented in a table or plot you should not repeat it again in the text. You can refer to and discuss general trends and perhaps even present average values as long as they are not already given in the table or plot.

4) Be sure to go over my comments in the attached document regarding the dendrogram and how it could be improved. This I think is the most important aspect of your study.

Response 4: We couldn't see the comments you mention in the document in this section.

5) The species of Stomatolepas you list is S. elegans. The species Stomatolepas praegustator has only been described from the mouth of sea turtles, Stomatolepas on the necks of olive ridleys are the species S. elegans (see Robinson, N.J., Lazo-Wasem, E., Butler, B.O., Lazo-Wasem, E.A., Zardus, J.D. & Pinou, T. (2019). Spatial distribution of epibionts on olive ridley sea turtles at playa ostional, costa rica. PLoS ONE14, e0218838).

Other studies have also found that Stomatolepas praegustator is genetically identical to S. elegans so the two are really the same species and the name S. elegans has priority (see Pinou, T., Lazo-Wasem, E.A., Dion, K. & Zardus, J.D. (2013). Six degrees of separation in barnacles? Assessing genetic variability in the sea-turtle epibiont stomatolepas elegans (costa) among turtles, beaches, and oceans. Journal of Natural History47, 2193-2212).

Response 5: we modify the species correctly and add the recommended reference

6) On line 216 you discuss Stephanolepas muricata and mention it was found in the neck. Is that right? This is not common so it is important to confirm. This barnacle is usually found only in the flipper.

Response 6: Yes, it is correct that we found the species S. muricata in the neck, even though other authors do not mention finding it in that part, if they mention that it adheres to the soft tissues and the neck is one of them.

7) On line 255 you discuss how Stephanolepas invades the skin and gets surrounded by a layer of fat (??) and you mention it affecting turtle eggs (??). I think this is in error. Please clarify or delete.

Response 7: Sorry, it was a writing error that has already been modified

8) In the discussion you could add more discussion regarding succession in epibiont communities on turtles since little has been studied. Below are the only two references that I know of on the topic.

Frick, M.G., Williams, K.L., Veljacic, D.C., Jackson, J.A. & Knight, S.E. (2002). Epibiont community succession on nesting loggerhead sea turtles, Caretta caretta, from Georgia, USA. In: Proceedings of the 20th Annual Symposium on Sea Turtle Biology and Conservation (eds. A. Mosier, A. Foley and B. Brost), pp. 281-282. NOAA technical memorandum NMFS-SEFSC-477.

Nolte, C.R., Pfaff, M.C., De Lecea, A.M., Le Gouvello, D. & Nel, R. (2020). Stable isotopes and epibiont communities reveal foraging habitats of nesting loggerhead turtles in the south West Indian Ocean. Marine Biology167, article 162.

Response 8: we attended to this observation and added to literature.

Kind regards

Reviewer 4 Report

The manuscript "Diversity of epibionts associated with Lepidochelys olivacea (Eschscholtz 1829) sea turtles nesting in the Mexican South Pacific" describes macrofaunal assemblages found on 43 turtles from 2017, including species richness, abundance, and distribution patterns.

The Introduction needs more in-depth descriptions of the current state of the literature and the gaps in said literature that this work is seeking to fill. A significant contribution that is missing in the manuscript is that of Ingels et al. (2020) Meiofauna life on loggerhead sea turtles - Diversely structured abundance and biodiversity hotspots that challenge the meiofauna paradox, and the references therein. Additional references are needed to support various statements throughout the Introduction and Discussion. The manuscript also needs to make clear from the beginning that this work focuses on macrofauna, particularly since most of the literature includes much smaller organisms.

Statistical assessment of the results can be improved, such as using a linear model instead of U Test for the relationship between epibiont abundance and turtle size or a chi squared test rather than a dendrogram for patterns between epibiont abundance and sampling month. I also have significant concerns regarding the need for, and validity of, such a temporal analysis. The manuscript describes 43 turtles from 5 months within a single year. Is this a sufficient sample size to evaluation a "month" effect, particularly considering one of the clusters includes the only 2 remoras found - which is more likely an outlier for nesting turtles. Are environmental or other conditions sufficiently different between months to necessitate such an analysis? In addition, why was the cloaca considered its own section given that the authors were focusing on macrofauna the same size as, or larger than, this region? When working under such constraints, it is no surprise that the authors found no epibionts here. 

The Discussion needs more description of the consequences of this work (i.e., why is this work so critical to justify someone's time reading this paper?). There are several statements in particular which need more work. For example, the authors report a statistically significant difference of 0.5cm in carapace width between turtles with and without epibionts. Though this is statistically significant, is 0.5cm biologically relevant? The authors also state that their monthly analysis suggests different foraging or breeding sites/populations which use the nesting beach sampled. However, this is insufficient evidence to support such a statement unless there is additional corroboration (e.g., link macrofaunal assemblages back to specific foraging or breeding grounds, differences in stable isotope signatures or tagging data commensurate with differences in epibionts).

I suggest reconsideration after major revision assuming the authors can strengthen their case for why this work is important, justify their choice of statistics, and provide a better description of the current state of the literature.

Author Response

Response to Reviewer 4 Comments

The manuscript "Diversity of epibionts associated with Lepidochelys olivacea (Eschscholtz 1829) sea turtles nesting in the Mexican South Pacific" describes macrofaunal assemblages found on 43 turtles from 2017, including species richness, abundance, and distribution patterns.

The Introduction needs more in-depth descriptions of the current state of the literature and the gaps in said literature that this work is seeking to fill. A significant contribution that is missing in the manuscript is that of Ingels et al. (2020) Meiofauna life on loggerhead sea turtles - Diversely structured abundance and biodiversity hotspots that challenge the meiofauna paradox, and the references therein. Additional references are needed to support various statements throughout the Introduction and Discussion. The manuscript also needs to make clear from the beginning that this work focuses on macrofauna, particularly since most of the literature includes much smaller organisms.

Response 1:  thank you for your comments, we listened to your suggestion and added the recommended reference

Statistical assessment of the results can be improved, such as using a linear model instead of U Test for the relationship between epibiont abundance and turtle size or a chi squared test rather than a dendrogram for patterns between epibiont abundance and sampling month. I also have significant concerns regarding the need for, and validity of, such a temporal analysis.

Response 2:  We maintain the same statistical tests by virtue of the fact that there is no mention that they are misapplied, however, we welcome your suggestions, we consider that these are correct and are useful for our study.

The manuscript describes 43 turtles from 5 months within a single year. Is this a sufficient sample size to evaluation a "month" effect, particularly considering one of the clusters includes the only 2 remoras found - which is more likely an outlier for nesting turtles. Are environmental or other conditions sufficiently different between months to necessitate such an analysis?

Response 3:  We found 43 turtles with epibionts, however, there were 125 observed turtles, the cluster was made with the information obtained, unfortunately, only 2 remoras were collected in the time that the investigation lasted. The study was carried out in those months because it is the nesting season for the olive ridley sea turtle and it was our opportunity to collect the epibionts.

In addition, why was the cloaca considered its own section given that the authors were focusing on macrofauna the same size as, or larger than, this region? When working under such constraints, it is no surprise that the authors found no epibionts here. 

Response 4: We considered this section in our study, because different authors have found epibionts housed in the turtle's cloaca, unfortunately, in our study we did not find organisms in this section

The Discussion needs more description of the consequences of this work (i.e., why is this work so critical to justify someone's time reading this paper?). There are several statements in particular which need more work. For example, the authors report a statistically significant difference of 0.5cm in carapace width between turtles with and without epibionts. Though this is statistically significant, is 0.5cm biologically relevant?

Response 5: When we speak of a significant difference, we provide evidence that reinforces the assumption that the older and wider the turtle, the greater its load of epibionts, which is expressed in the writing.

The authors also state that their monthly analysis suggests different foraging or breeding sites/populations which use the nesting beach sampled. However, this is insufficient evidence to support such a statement unless there is additional corroboration (e.g., link macrofaunal assemblages back to specific foraging or breeding grounds, differences in stable isotope signatures or tagging data commensurate with differences in epibionts).

Response 6: The cluster analysis gave us the similarity of the presence of certain species in a certain time or months, so we concluded that the turtles that presented this species similarity could have the same migration zones. in this study, stable isotope analysis was not used.

I suggest reconsideration after major revision assuming the authors can strengthen their case for why this work is important, justify their choice of statistics, and provide a better description of the current state of the literature.

Response 7: we attend to your suggestions and corrections

Round 2

Reviewer 4 Report

Thank you for your responses to my concerns within the initial submission. Though I:

1) still believe the authors are overstating the biological relevance of the 0.5 cm difference in curved width,

2) would have greater confidence in the monthly analysis if it included multiple years to ensure it was not simply an isolated pattern unique to 2017, and

3) would prefer to see more detail presented in the Introduction and Discussion to put the manuscript in a more global context and highlight the importance of this work's findings,

I do not feel that these hesitations should prohibit the publication of these findings as they may be useful for local management. Grammar revisions are still needed throughout the manuscript, particularly related to run-on sentences. After such grammatical revisions, I would suggest this manuscript's acceptance for publication.

Author Response

Point 1.

1) still believe the authors are overstating the biological relevance of the 0.5 cm difference in curved width,

Response 1

There are very few studies that relate age with morphometric measurements of the sea turtle. But according to Zug et al. 2006 (Figure 3), we can see that the turtle reaches a maximum length at 15 years; from there, the growth is lower and constant; therefore, the difference of 2.5 cm found in this study could indicate >10 years by Zug et al. 2006, and some marine turtles can live more than 50 years, which in biological and ecological terms are very important for the turtle.

Zug GR, Chaloupka M, Balazs GH (2006) Age and growth in olive ridley seaturtles (Lepidochelys olivacea) from the North-central Pacific: a skelotochronogical analysis. Marine Ecology, 27: 263-270.

Point 2.

2) would have greater confidence in the monthly analysis if it included multiple years to ensure it was not simply an isolated pattern unique to 2017, and

Response 2

Multivariate analyzes such as clusters are always exploratory and descriptive and should be interpreted that way. We observed from the analysis that probably different populations of turtles are arriving at Llano Real, but we agree that more data and studies will be required to corroborate this hypothesis and determine if epibionts can be used for this purpose. Therefore, this study proposes new hypotheses and future research avenues.

Point 3

3) would prefer to see more detail presented in the Introduction and Discussion to put the manuscript in a more global context and highlight the importance of this work's findings,

Response 3

The objective of this study was to fill a gap that existed in the south of the Mexican Pacific. On the other hand, there are too many studies in the north region (Hernández-Vázquez y Valadez-González 1998; Lazo-Wasen et al. 2007; Suárez-Morales et al. 2009; Lazo-Wasem et al. 2011;). It is a specific site, therefore, trying to infer more of the results is out of our reach; however, this study will lay the foundations to develop multidisciplinary studies that allow us to answer questions of greater biological and ecological relevance using our results.

Thank you for all your comments.
